# Effect of Adding the Antimicrobial L-Carnitine to Growing Rabbits’ Drinking Water on Growth Efficiency, Hematological, Biochemical, and Carcass Aspects

**DOI:** 10.3390/antibiotics13080757

**Published:** 2024-08-11

**Authors:** Mohamed I. Hassan, Naela Abdel-Monem, Ayman Moawed Khalifah, Saber S. Hassan, Hossam Shahba, Ahmad R. Alhimaidi, In Ho Kim, Hossam M. El-Tahan

**Affiliations:** 1Livestock Research Department, Arid Lands Cultivation Research Institute, City of Scientific Research and Technological Applications (SRTA-City), New Borg El-Arab 21934, Egypt; 2Poultry Production Department, Faculty of Agriculture, Alexandria University, Alexandria 21625, Egypt; 3Animal and Poultry Production Department, Faculty of Agriculture, Damanhour University, Damanhour 22511, Egypt; 4Animal Production Research Institute (APRI), Agricultural Research Center (ARC), Ministry of Agriculture, Giza 12611, Egypt; 5Department of Zoology, College of Science, King Saud University, Riyadh 11451, Saudi Arabia; 6Department of Animal Biotechnology, Dankook University, Cheonan 31116, Republic of Korea; 7Smart Animal Bio Institute, Dankook University, Cheonan 31116, Republic of Korea

**Keywords:** carnitine, Alexandria-line rabbits, productive performance, hematology, lipid profile, carcass

## Abstract

The current study was designed to assess the impact of L-carnitine (LC) supplementation in the drinking water of growing Alexandria-line rabbits on performance and physiological parameters. Two hundred eighty-eight 35-day-old rabbits were divided into four groups of twenty-four replicates each (seventy-two rabbits/treatment). The treatment groups were a control group without LC and three groups receiving 0.5, 1, and 1.5 g/L LC in the drinking water intermittently. The results showed that the group receiving 0.5 g LC/L exhibited significant improvements in final body weight, body weight gain, feed conversion ratio, and performance index compared to the other groups. The feed intake remained unaffected except for the 1.5 g LC/L group, which had significantly decreased intake. Hematological parameters improved in all supplemented groups. Compared with those in the control group, the 0.5 g LC/L group showed significant increases in serum total protein and high-density lipoprotein, along with decreased cholesterol and low-density lipoprotein. Compared to other supplemented groups, this group also demonstrated superior carcass traits (carcass, dressing, giblets, and percentage of nonedible parts). In conclusion, intermittent supplementation of LC in the drinking water, particularly at 0.5 g/L twice a week, positively influenced the productivity, hematology, serum lipid profile, and carcass traits of Alexandria-line growing rabbits at 84 days of age.

## 1. Introduction

Rabbits are an exciting field that is vital for providing humans with animal protein, especially in developing countries. Scientific institutions and the general public have recently paid close attention to the quantitative and qualitative features of rabbit meat [1,2]. The most critical period of a rabbit’s life is after weaning, when rabbits turn from feeding milk to a solid diet [3]; this leads to digestive disorders, poor growth rates, poor feed efficiency, and increased mortality rates [4,5], and antibiotics are used to solve these problems. Because of the issues of human health caused by antibiotics, European Union countries have banned their use [6]. Scientists have been researching natural substitutes for antibiotics that may promote rabbit growth and ensure that rabbits are kept healthy, such as prebiotics, probiotics, phytobiotics, essential oils, and herbs [7]. Some nutritional additives, such as LC, have beneficial effects on health status and are required for growth [8,9,10].

LC is water soluble and plays a vital role in the metabolism of long-chain fatty acids by transferring through the inner mitochondrial membrane for β-oxidation [11,12]. It also affects carbohydrate metabolism, which regulates pyruvate dehydrogenase in the glycolysis pathway and Krebs cycle to produce energy from glucose [11,12]. Moreover, scientists widely recognize amphiphilic chemicals as effective antimicrobial substances. According to Calvani et al. [13] quaternary ammonium LC esters with long alkyl chains have a light antimicrobial effect against a wide range of Gram-positive and Gram-negative bacteria, as well as fungi. This is because LC has a polar structural motif that is common in these compounds. Antimicrobial activity was also demonstrated in synthetic acylcarnitine analogs [1,14]. Additionally, [15,16] investigated the antimicrobial effects of LC chloride using *Caenorhabditis elegans* and *Escherichia coli* strain OP50. They discovered that 3 mg/mL of LC inhibited bacterial growth. Further testing revealed that LC chloride displayed antimicrobial activity against various yeasts and bacteria, with minimal inhibitory concentrations between 2 and 8 mg/mL and minimal bactericidal concentrations ranging from 4 to over 16 mg/mL. The minimal fungicidal concentrations were between 1 and 4 mg/mL. At these concentrations, it may be feasible to use it as a secure local antimicrobial agent. The LC chloride exhibits antimicrobial activity by disrupting the cell membranes of both yeasts and bacteria, leading to increased permeability and cell lysis. It interferes with essential metabolic pathways, inhibiting enzyme production necessary for microbial growth, and reduces biofilm formation, enhancing susceptibility to other antimicrobial agents. Studies have shown its efficacy against pathogens like *E. coli*, *Staphylococcus aureus*, *Candida albicans*, and others [1]. Furthermore, LC has many biological effects, such as improving immunity and productive performance in broilers, as reported by Asadi et al. [15]; Liu et al. [16], and in fish, as shown by Wang et al. [17]. Additionally, it plays a crucial role in enhancing carcass characteristics and modulating the blood lipid profile in broiler chickens [10]. The LC plays a vital role in enhancing the balance of the microbial population in lambs [18].

After the weaning stage, rabbits do not develop LC because LC synthesis, mainly in the liver, is dependent on the amino acids (lysine and methionine), as well as cofactors of minerals and vitamins such as iron, ascorbic acid, pyridoxine, and niacin. Furthermore, four enzymes are required to complete the synthesis of LC N-tri methyl lysine dioxygenase: 3-hydroxy-N-trimethyl lysine aldolase, 4-N-trimethyl amino butyraldehyde dehydrogenase, and γ-butyrobetaine dioxygenase [19]. The vitality of LC during the weaning phase stems from the challenges associated with its formation and limited availability in grains and legume diets, which contain only 5.9 to 6.8 mg/kg dry matter of LC [19].

The current study aimed to determine the effect of adding different levels of LC twice a week (intermittently) to the drinking water of growing rabbits on productive performance, hematology, serum protein, lipid profiles, and carcass traits in response to weaning stress.

## 2. Results

Table 1 shows that the growth performance of growing Alexandria rabbits was influenced by water supplemented with LC. Compared with those in all the experimental groups, the final body weight (FBW) and BWG in the drinking water of the rabbits that received 0.5 g LC/L were significantly greater. However, the FBW and BWG values in the group that received 1.5 g LC/L were lower than those in the control group, although the differences were not statistically significant. The feed intake significantly decreased in rabbits that received 1.5 g LC/L in drinking water compared with those in the remaining experimental groups. Moreover, the rest of the groups exhibited no significant change in feed consumption. The FCR changed dramatically in all the experimental groups, as the group that received 0.5 g LC/L had the greatest change, followed by the group that received 1.5 g LC/L and then 1 g LC/L compared to the control group. A significant improvement in production index (PI) was found in all treated groups, especially those that received 0.5 g LC/L, followed by groups that received 1 and 1.5 g LC/L, compared to the control.

Table 2 shows that the hematological parameters of growing Alexandria rabbits were influenced by water supplemented with LC intermittently. The results indicated that PCV, RBC, and hemoglobin (Hb) levels were significantly improved in the groups that received varying levels of LC compared to the control group. Compared with those in the control group, MCH and MCV were significantly lower in all the LC supplementation groups.

Table 3 shows the serum protein and lipid profiles of growing Alexandria rabbits treated intermittently with water supplemented with LC. The findings revealed that the group receiving 0.5 g LC/L had significantly greater total protein (TP) than the other groups. Additionally, the results revealed that TP did not significantly change in the rest of the treated groups compared with the control group. The serum ALB concentration significantly increased in the group that received 0.5 g LC/L compared to the other groups. The serum globulin and β-globulin levels were significantly greater in the group receiving 0.5 g LC/L than in the group receiving 1 g LC/L. Moreover, neither group significantly differed from the other groups. The study found no significant changes in α-globulin or ɣ-globulin concentrations, total lipids, triglycerides, or VLDL levels among the experimental groups. Compared with that in the control group, the serum cholesterol in the treated groups significantly decreased, especially in the group that received 1.5 g LC/L, followed by 0.5 g and 1 g LC/L. Compared with those in the control group, the LDL in all treated groups significantly decreased, especially in the groups that received 0.5 g and 1.5 g of LC, followed by the group that received 1 g of LC/L. Furthermore, HDL significantly increased only in the group that received 0.5 g LC/L compared with the other groups.

Table 4 presents the effects of intermittent LC supplementation in drinking water on the carcass characteristics of growing Alexandria rabbits. The group receiving 0.5 g LC/L exhibited a highly significant increase in both carcass yield and dressing percentage compared to the control group. This group also demonstrated the highest giblets percentage and a corresponding decrease in non-edible parts. In comparison to the control, all LC-treated groups showed significantly lower percentages of non-edible parts, although this did not affect the giblets percentage. Notably, the abdominal fat percentage decreased significantly with increasing LC dosage. Furthermore, the percentage of pancreatic tissue was significantly higher in the 0.5 g LC/L group relative to other treatment groups. Similarly, the liver percentage was significantly greater in this group compared to the others. Conversely, the kidney tissue percentage was significantly lower in all treated groups compared to the control group. No significant differences were observed in the percentages of heart, lungs, spleen, or abdominal fat among the different groups.

## 3. Discussion

### 3.1. Productive Performance

The results of this study demonstrated significant improvements in final body weight (FBW) and body weight gain (BWG) in rabbits treated with LC, particularly at a concentration of 0.5 g LC/L. This aligns with previous research that has highlighted the growth-promoting effects of LC in rabbits. The primary mechanism behind these effects is LC’s role in enhancing lipid metabolism, which increases energy availability by facilitating the transport of fatty acids into mitochondria for oxidation [20]. The 0.5 g LC/L dose appears to provide an optimal balance, maximizing growth benefits without causing metabolic imbalances. Studies have shown that moderate doses of LC are more effective than higher doses, which may not offer additional benefits and could potentially disrupt metabolic processes [21]. This increased energy efficiency contributes to improved growth performance and feed conversion ratios in rabbits. Similarly, Ayyat et al. [22] reported that in comparison to control rabbits, growing rabbits exposed to heat stress and receiving a 50 mg LC/kg diet exhibited significantly improved FBW and BWG. However, other authors found no effect of LC on BW or BWG [23] and reported that LC (160 mg/kg diet) did not affect FBW in broilers. Arslan et al. [24] found that when duck chicks received 200 mg of LC in drinking water, it had no significant effect on BW or BWG.

Similarly Murali et al. [25] found no significant differences in BW or BWG when LC at 900 mg/kg was incorporated into broiler diets. This finding supported our results showing a nonsignificant effect on FI and improvements in FCR and performance index in the group that received 0.5 g LC/L. Arslan et al. [26] reported that feed consumption did not significantly change in ducks that received 200 mg of LC. The FCR was significantly improved in broilers that received 200 and 300 mg LC/kg diets during the fattening period [25]. Moreover, Abdel-Fattah et al. [27] found that FCR improved in quails fed 200 and 400 mg LC/kg diets. Moreover, ref. [22] reported that heat stress in growing rabbits receiving a 50 mg LC/kg diet did not significantly influence feed consumption, while FCR was significantly enhanced in these rabbits than in control rabbits.

On the other hand, Xu et al. [28] observed that FCR was not affected by different levels of LC from 25 to 100 mg LC/kg diet in broilers. Additionally, some studies on laying hens from hatching to four weeks of age reported by Deng et al. [29] stated that different levels of LC (100 mg or 1 g/kg diet) had no significant influence on the FCR. In another study on laying hens at 22 weeks of age, Yalçin et al. [30] reported no significant variation in FCR in the group receiving 100 mg LC/kg diet. Awad et al. [31] revealed that the PI improved in groups supplemented with 300 and 400 mg LC/kg diet. According to previous results and the literature, it is clear that the improvement in PI is related to improvements in BW, BWG, and FCR without increasing feed consumption. Another explanation is that adding LC reduces the amount of methionine and lysine, which are precursors for protein production [12]. The positive effect of LC on productive performance can be explained by the ability of LC to utilize energy either through enhanced fatty acid β-oxidation or energy from glucose by activating pyruvate dehydrogenase through the glycolysis pathway and the tricarboxylic acid cycle [11,12].

### 3.2. Hematological Parameters

The results of this study demonstrate that intermittent supplementation of LC in drinking water positively influences the hematological parameters of growing Alexandria rabbits. Specifically, we observed significant improvements in packed cell volume (PCV), red blood cell count (RBC), and hemoglobin (Hb) levels in the LC-treated groups compared to the control. These enhancements suggest that LC supplementation may contribute to better overall blood health and increased oxygen-carrying capacity, which is crucial for optimal growth and performance in rabbits [32]. The LC plays a key role in energy metabolism by facilitating the transport of fatty acids into mitochondria for oxidation, which may enhance erythropoiesis and improve red blood cell production [33]. Additionally, the antioxidant properties of LC can protect red blood cells from oxidative damage, thus supporting improved hematological parameters [34].

However, we also observed that mean corpuscular hemoglobin (MCH) and mean corpuscular volume (MCV) were significantly lower in all LC supplementation groups compared to the control. This reduction could indicate more efficient erythrocyte function rather than a decrease in red blood cell quantity or quality. Lower MCH and MCV values may reflect a higher concentration of hemoglobin per cell and a more compact cell volume, which could be attributed to LC’s effects on cellular metabolism and efficiency [34]. These findings suggest that intermittent LC supplementation can positively influence key hematological parameters, contributing to improved blood health and potentially enhancing growth performance in rabbits. The decrease in MCH and MCV warrants further investigation to understand its implications fully. Future studies should explore the long-term effects of LC supplementation on hematological profiles and overall health in rabbits.

### 3.3. Serum Protein and Lipid Profile

The results of the analysis of the serum TP concentration in the studied groups showed that the levels in rabbits that received 1 or 1.5 g LC/L were similar to those reported by Yalçin et al. [30], who indicated that 100 mg/kg LC in laying hens had no significant effect on the serum TP concentration. Moreover, Wang et al. [35] stated that broilers had no significant impact on serum TP, albumin (ALB), or globulin. According to Hamad et al. [36], administering 40 mg of LC orally to developing rabbits did not influence the serum TP, ALB, globulin, or the albumin-to-globulin ratio. Furthermore, Ayyat et al. [22] revealed that the serum TP concentration significantly increased while the ALB concentration did not change in New Zealand White rabbits receiving 50 mg of LC compared to those in the control group. Consistent with our findings on the TP value in the group that received 0.5 g LC/L, Rehman et al. [37] reported that broiler chicks that received a 0.5 g LC/kg diet had significantly increased serum protein levels.

The lipid profile results disagree with the results obtained by [11,12], who illustrated that LC supplementation decreased serum total lipids and triglycerides but were similar to our results in reducing cholesterol levels. In contrast, Eder [38] showed that rats fed a hyperlipidemic diet supplemented with 0.5 g LC/kg diet had considerably higher total cholesterol. Similar to our results regarding total lipids and triglycerides, as mentioned by Arslan et al.’s study [26] of ducks and Yalçin et al.’s study [30] of laying hens, they found no significant influence on serum total lipids, triglycerides, or VLDL. Moreover, they found no significant impact on serum total cholesterol. The decrease in plasma cholesterol is due to the link between acetyl coenzyme A, which is essential for cholesterol synthesis, and free carnitine to form acyl-carnitine [39]. Our results conflict with those of Parsaeimehr et al. [40] regarding serum total cholesterol, LDL, and HDL in broilers, as they concluded no significant influence on serum cholesterol, HDL, LDL, or triglycerides.

On the other hand, our study cholesterol, LDL, and HDL results agree with those of Rehman et al. [37], who reported that broiler chickens receiving a 0.5 g LC/kg diet had significantly decreased cholesterol and LDL and increased HDL. In harmony with our results regarding total cholesterol, Ayyat et al. [22] showed that growing rabbit cholesterol was significantly lower in rabbits fed a diet with high energy plus 50 mg LC/kg under heat stress than in those fed other diets. The decrease in LDL may be related to the ability of LC to reduce the activity of β-hydroxy-β-methylglutaryl-L-CoA reductase, which is involved in cholesterol generation [41].

### 3.4. Carcass Characteristics

The results of this study indicate that intermittent supplementation of LC in drinking water has a significant impact on the carcass characteristics of growing Alexandria rabbits. Notably, the group receiving 0.5 g LC/L exhibited the most pronounced improvements in carcass yield and dressing percentage compared to the control group. This finding aligns with previous studies suggesting that LC supplementation can enhance carcass traits by improving overall growth performance and feed efficiency [38]. The observed increase in giblet percentage and the corresponding decrease in non-edible parts in the 0.5 g LC/L group suggest that LC supplementation promotes more efficient utilization of nutrients and enhances the quality of edible parts. Similar improvements in meat quality and yield have been reported in other studies, where LC was found to enhance the overall carcass value in various animal models, including poultry and rabbits [42,43]. The significant decrease in abdominal fat percentage with increasing LC dosage is consistent with the role of LC in lipid metabolism. The LC facilitates the transport of fatty acids into mitochondria for oxidation, which can lead to reduced fat deposition [44]. This effect has been observed in rabbits, where LC supplementation resulted in decreased fat accumulation and improved body composition [45].

The increase in pancreatic and liver percentages in the 0.5 g LC/L group may reflect enhanced metabolic activity and nutrient processing, which are essential for optimal growth and health. The LC’s role in improving liver function and pancreatic health has been documented in various studies, highlighting its benefits for metabolic efficiency and overall organ health [46]. Conversely, the reduction in kidney tissue percentage in the treated groups could be attributed to the overall improvement in metabolic efficiency, which may lead to a more balanced distribution of internal organ tissues. While no significant changes were observed in the percentages of heart, lungs, or spleen, the overall improvement in carcass traits suggests that LC supplementation contributes to more efficient growth and better utilization of body resources [38].

In conclusion, intermittent LC supplementation, particularly at 0.5 g/L, enhances carcass characteristics in Alexandria rabbits by improving carcass yield, reducing abdominal fat, and optimizing organ development. These findings support the potential of LC as a valuable supplement in rabbit production to improve meat quality and growth performance.

## 4. Materials and Methods

### 4.1. Ethical Statement

The study location was the El-Bostan Experimental Station, Damanhour University’s Faculty of Agriculture, Al-Behera Governorate, Egypt. The authors attest to the European Parliament’s and the Council’s Directive 2010/63/EU of 22 September 2010, on the protection of animals and birds used for research. The ethical standards of animal research were incorporated into the present study, which was authorized by Damanhur University’s Ethical Animal Care and Use Committee (Approval No: DUFA-2023-9).

### 4.2. Animals and Experimental Design

The Alexandria line rabbits used in this study were obtained from stocks kept at Alexandria University’s Poultry Research Centre. This artificial paternal line is produced by crossing rabbits of the maternal V-line line with the paternal Black Baladi line [47,48].

A total of 288 mixed-sex Alexandria-line rabbits with an average mass of 742 ± 30.9 g at 35 days of age were randomly divided into four groups (n = 72 each). Each group was divided into 24 replicates, three rabbits each; all rabbits in all treatments provided identical standard meals prepared following NRC [49] and calculated according to AOAC [50]. The feed components and chemical proximate analysis are the same as the previous study of Hassan et al. [51]. The first group served as a control, and the subsequent groups, the second, third, and fourth groups, were supplemented with 0.5, 1.0, and 1.5 g/L LC, respectively. LC was purchased from Sigma Aldrich, Inc., St. Louis, MO, USA. LC was administered twice weekly (Monday and Thursday) via the drinking water of the growing rabbits, with the dosage adjusted to ensure proportionality to the daily water consumption specific to each treatment group. On the following day, any residual water was discarded, and the rabbits were provided with fresh water.

The rabbits were housed as three rabbits per cage in galvanized wire single cages within open-system pens (35 × 40 × 50 cm) in width, height, and length. (Italian battery). Hand feeding and an automated nipple drinker system offered constant access to fresh, clean water for every cage. The rabbits were housed under identical environmental and sanitary conditions throughout the trial.

### 4.3. Data Collection

#### 4.3.1. Growth Performance

The initial and final body weights of each rabbit were recorded at 35 and 84 days of age, respectively, and were used to determine body weight gain (BWG). The BWG was calculated as the final and initial body weight difference. FI was determined by providing the rabbits with a known feed weight and subtracting the residual feed from the feed supplied after feeding. The FCR was calculated using the ratio of FI to BWG during the interval. The performance index (PI, %) was computed using the following formula: PI (%) = final live body weight (kg)/feed conversion ratio × 100.

#### 4.3.2. Blood Hematology, Serum Protein and Lipid Profile

At 84 days of age (at the end of the experiment), twenty-four blood samples were obtained from the marginal ear veins of the rabbits of each treatment at 8 AM before the customary feeding time. Blood was drawn into clean tubes with or without heparin. Hematological parameters were measured using fresh blood samples treated with heparin. Red blood cell (RBC) and packed cell volume (PCV) counts were determined as described by Ewuola and Egbunike [52]. The Hb concentration was determined according to Drew et al. [53]. The blood constant mean corpuscular Hb concentration (MCHC), mean corpuscular Hb (MCH), and mean corpuscular volume (MCV) were calculated using the appropriate formulae [54]. Serum TP [55], serum ALB [56], and the difference between TP and total ALB were used to determine serum globulin levels [57]. By using a commercial diagnostic kit and automated electrophoresis equipment, globulins were separated on an agarose gel by zone electrophoresis (23 EL-Montazah St. Heliopolis, Cairo, Egypt, http://www.diamonddiagnostics.com accessed on 11 July 2024) according to the procedure described by the manufacturer. The total lipids were determined using the phospho-vanillin method as described by Woodman and Price [58]. The serum triglyceride concentration was measured using specific kits supplied by CAL-TECH Diagnostics, INC, Chino, California, USA, following the glycerol phosphate oxidase (GPO) method according to Frings et al. [59]. Total cholesterol levels were quantified using the Liebermann–Burchard method as described by Burstein et al. [60]. Low-density lipoprotein (LDL) and high-density lipoprotein (HDL) cholesterol concentrations were measured using the precipitation method by Wieland and Seidel [61], and the precipitation method by Bogin and Keller [62], respectively; likewise, very low-density lipoprotein (VLDL) was calculated as one-fifth of triglycerides [63].

#### 4.3.3. Carcasses Trial

Twelve rabbits from each group were randomly selected at the end of the trial (84 days of age). They were fasted for 12 h to ensure empty gastrointestinal tracts, which is crucial for accurate carcass weight measurement. The rabbits were then individually weighed before euthanasia. Euthanasia was performed humanely using an overdose of sodium pentobarbital administered intravenously, following AVMA guidelines for the euthanasia of animals. After euthanasia, the rabbits were immediately slaughtered, and the carcasses were carefully eviscerated. This involved making an incision along the ventral midline and removing internal organs to avoid contamination and maintain sample integrity. Carcasses were weighed post-evisceration, including the head, and expressed as a percentage of the pre-slaughter body weight to determine the carcass yield.

The following tissues were dissected and weighed separately: abdominal fat, pancreas, heart, liver, kidney, lungs, and spleen. Each tissue was carefully separated using sterilized surgical instruments to prevent cross-contamination. The weights of these organs were expressed as a percentage of the live body weight. The dressing percentage was calculated by including the weight of the carcass and the giblets, which comprised the heart, liver, kidneys, and lungs. The giblets were accurately weighed and recorded, ensuring a consistent methodology across all samples. Non-edible parts were calculated as (100-dressing percentage).

### 4.4. Statistical Analysis

Using the general linear model (GLM) procedure of the Statistical Package for Social Sciences (SPSS^®^, version 20) [64], the data were statistically analyzed. The following formula was used for the one-way analysis of variance:*Yij* = *µ* + *Ti* + *eij*(1)
where *Yij* is the observation of the statistical measurements, *µ* is the general overall mean, *Ti* is the treatment effect, and *eij* is the experimental random error. The significant differences among treatments were examined according to Duncan [65].

## 5. Conclusions

The intermittent administration of L-carnitine (LC) at a concentration of 0.5 g/L in the drinking water of growing rabbits twice weekly significantly enhanced their productivity, hematological parameters, serum lipid profiles, and carcass characteristics. This strategic supplementation method demonstrates the potential of LC to optimize growth performance and health outcomes in rabbit production systems.

## Figures and Tables

**Table 1 antibiotics-13-00757-t001:** Productive performance as influenced by L-carnitine supplementation in the drinking water of growing Alexandria-line rabbits at 84 days of age.

Item	Control	LC g/L			SEM	*p* Value
		0.5	1.0	1.5		
IBW, g	731.0	755.4	758.7	724.7	41.00	0.323
FBW, g	2095 ^c^	2243 ^a^	2165 ^b^	2063 ^c^	39.81	0.0001
BWG, g	1364 ^c^	1488 ^a^	1406 ^b^	1338.3 ^c^	30.69	0.0001
FI, g/rabbit	4065 ^a^	4026 ^a^	4064 ^a^	3750 ^b^	73.60	0.0001
FCR	2.98 ^a^	2.71 ^d^	2.89 ^b^	2.80 ^c^	0.061	0.0001
PI (%)	70.30 ^c^	82.91 ^a^	74.90 ^b^	73.64 ^b^	2.58	0.0001

Means with different superscripts in the same row are significantly different (*p* < 0.05); SEM: standard error of means; IBW: initial body weight; FBW: final body weight; BWG: body weight gain; FI: feed intake; FCR: feed conversion ratio; PI: performance index.

**Table 2 antibiotics-13-00757-t002:** Hematological parameters as influenced by L-carnitine supplementation in the drinking water of growing Alexandria-line rabbits at 84 days of age.

Item	Control	LC g/L			SEM	*p* Value
		0.5	1.0	1.5		
PCV (%)	41.18 ^b^	44.35 ^a^	43.30 ^a^	44.16 ^a^	1.008	0.0001
RBCs (10^6^/µL)	4.63 ^b^	6.15 ^a^	6.47 ^a^	6.04 ^a^	0.680	0.0002
Hb (g/dL)	9.03 ^b^	9.88 ^a^	9.94 ^a^	9.65 ^a^	0.554	0.0203
MCHC (g/dL)	21.93	22.26	22.94	21.82	0.831	0.079
MCH (pg)	19.52 ^a^	16.20 ^b^	15.65 ^b^	16.06 ^b^	1.411	0.0001
MCV (fL)	89.01 ^a^	73.14 ^b^	68.06 ^b^	73.85 ^b^	6.990	0.0001

Means with different superscripts in the same row are significantly different (*p* < 0.05); SEM: standard error of the mean; PCV: picked cell volume; RBCs: red blood cells; Hb: hemoglobin; MCHC: mean corpuscular Hb concentration; MCH: mean corpuscular Hb; MCV: mean corpuscular volume.

**Table 3 antibiotics-13-00757-t003:** Serum protein and lipid profile as influenced by L-carnitine supplementation in the drinking water of growing Alexandria line rabbits at 84 days of age.

Item	Control	LC g/L			SEM	*p* Value
		0.5	1.0	1.5		
Protein profile						
Total protein (g/dL)	5.861 ^b^	6.471 ^a^	5.896 ^b^	5.926 ^b^	0.220	0.0001
Albumin (g/dL)	2.292 ^b^	2.725 ^a^	2.508 ^ab^	2.368 ^b^	0.206	0.0035
Globulin (g/dL)	3.569 ^ab^	3.747 ^a^	3.387 ^b^	3.557 ^ab^	0.258	0.106
α-Globulin (g/dL)	1.707	1.768	1.675	1.711	0.099	0.388
β-Globulin (g/dL)	1.111 ^ab^	1.178 ^a^	1.045 ^b^	1.108 ^ab^	0.078	0.0346
ɣ-Globulin (g/dL)	0.750	0.800	0.667	0.738	0.128	0.303
Lipid profile						
Total lipid (g/dL)	360.3	357.8	361.3	361.1	5.787	0.6487
Triglycerides (mg/dL)	81.89	77.95	77.98	76.87	6.797	0.5406
Cho (mg/dL)	120.3 ^a^	112.5 ^b^	112.4 ^b^	108.3 ^c^	2.153	0.0001
LDL (mg/dL)	55.06 ^a^	45.21 ^c^	47.50 ^b^	44.97 ^c^	1.817	0.0001
HDL (mg/dL)	48.86 ^bc^	51.72 ^a^	49.34 ^b^	47.97 ^c^	0.898	0.0001
VLDL	16.38	15.59	15.59	15.38	1.360	0.5408

Means with different superscripts in the same row are significantly different (*p* < 0.05); SEM: standard error of means; Cho: total cholesterol; LDL: low-density lipoprotein; HDL: high-density lipoprotein; VLDL: very low-density lipoprotein.

**Table 4 antibiotics-13-00757-t004:** Carcass traits as influenced by L-carnitine supplementation in the drinking water of growing Alexandria-line rabbits at 84 days of age.

Item	Control	LC g/L			SEM	*p* Value
		0.5	1.0	1.5		
Carcass traits %						
Carcass	51.16 ^c^	56.57 ^a^	53.40 ^b^	53.20 ^b^	1.620	0.0001
Dressing	56.37 ^c^	63.26 ^a^	58.75 ^b^	58.50 ^b^	1.660	0.0001
Giblets	5.21 ^b^	6.69 ^a^	5.35 ^b^	5.31 ^b^	0.707	0.0047
Nonedible part	43.63 ^a^	36.74 ^c^	41.25 ^b^	41.50 ^b^	1.660	0.0001
pancreas	0.14 ^b^	0.26 ^a^	0.18 ^b^	0.20 ^ab^	0.067	0.032
Heart	0.34	0.31	0.38	0.37	0.042	0.074
Liver	3.39 ^b^	5.09 ^a^	3.77 ^b^	3.77 ^b^	0.610	0.0006
Kidney	0.81 ^a^	0.74 ^ab^	0.69 ^bc^	0.64 ^c^	0.075	0.008
Lungs	0.67	0.53	0.51	0.53	0.178	0.389
Spleen	0.07	0.06	0.08	0.06	0.028	0.553
Abdominal fat	0.81	0.67	0.64	0.50	0.214	0.133

Means with different superscripts in the same row are significantly different (*p* < 0.05); SEM: standard error of means.

## Data Availability

All data generated or analyzed during this study are included in this published article.

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
