# Peer review of "Effect of Adding the Antimicrobial L-Carnitine to Growing Rabbits’ Drinking Water on Growth Efficiency, Hematological, Biochemical, and Carcass Aspects"

_antibiotics, 2024, doi:10.3390/antibiotics13080757_

Round 1
Reviewer 1 Report
Comments and Suggestions for Authors
This is an interesting manuscript reporting results after the supplementation of drinking water of growing rabbits with L-Carnitine. In general, it is a well-written manuscript. The introduction is adequate and includes all the necessary information for the reader. The material and methods section presents the methods used clearly. The results are nicely presented and the discussion presents similar results from the relevant literature meaningfully. Below are a few recommendations that could improve the manuscript further and make it suitable for publication.
Introduction
Line 51: who are those "people"? you mean scientists? authors? please be more precise.
Line 57-62: This section needs rephrasing. It contains too much unrelated information. Please make it one sentence with only the vital information.
Line 66: What does "cultivating the carcass traits" mean? I do not understand. Please choose a different verb.
Line 69-73: Why do the rabbits not produce L-carnitine after weaning? You mention the factors that take part in the process but you do not state which is the limiting one. Is it the low intake of amino acids? the absence of the enzymes? Please clarify.
Results
Lines 85-87: when the difference is not significant, then the values are not greater or lower, they are the same. So please don't state that the values are greater or lower because it lacks any meaning. Rephrase.
Table 1: Are there really differences in FCR? How can values like 2.89 and 2.80 be different when the standard error is 0.06? Of course, the Duncan criterium is not very good in separating close values but please check the statistics one more time. Maybe using a more strict criterium like Tukeys of Bonferroni is more suitable.
Line 103:Meaningfully means nothing. It is either statistically significant or not. Change.
Line 104-107 and table 2: MCHC has a p-value of 0.079. Therefore there are no statistical differences here. So you need to revise the text and remove the superscripts from the table. This p-value indicates a tendency to differ (P<0.10) which if needed to be put on the table, should be with upper case letters (A, B etc.) and clearly indicated in the legend of the table.
Lines 123-125: Combine these two sentences in one since they say the same thing.
Discussion
Lines 209-211: What does fish oil has to do with this paper? Is it fish oil and L-carnitine? is fish oil rich in carnitine? Please clarify or remove.
Lines 222-225: Please avoid using "Moreover" too many times. Peek an alternative.
Materials and methods
You state that L-carnitine was added twice a week but the rabbits had access to water through nipple drinkers constantly. So what was the case: there was a water barrel where you added the carnitine, after calculating the amount required for 4 days? Or the carnitine was added at a standard volume and when the rabbits consumed that then they had access to just clean water? Please clarify if the animals consumed water supplemented with carnitine constantly. Here it would be helpful to have measured the amount of water consumed by the rabbits so that someone could calculate how much carnitine is needed to achieve such important results.
Author Response
Response to Reviewer 1:
Thank you for your review of our paper. We have answered your points below.
Response
Done as requested; line 1
Introduction
Line 51: who are those "people"? you mean scientists? authors? please be more precise.
Response
Done as requested. The word has been changed.
Line 57-62: This section needs rephrasing. It contains too much unrelated information. Please make it one sentence with only the vital information.
Response
Done as requested.
Line 66: What does "cultivating the carcass traits" mean? I do not understand. Please choose a different verb.
Response
Done as requested. This sentence has been rewritten.
Line 69-73: Why do the rabbits not produce L-carnitine after weaning? You mention the factors that take part in the process but you do not state which is the limiting one. Is it the low intake of amino acids? the absence of the enzymes? Please clarify.
Response
In fact, we looked at the literature and found that the combination of dietary changes, genetic predisposition, and physiological adaptations in rabbits after weaning leads to insufficient L-carnitine production. Supplementation with L-carnitine is often necessary to support optimal growth, health, and metabolic function in weaned rabbits, particularly in commercial settings where enhanced performance is desired (Mansoub, 2011; Cresci & Cannavino, 2013).
Mansoub, N. H. (2011). Effect of L-Carnitine Supplementation on Performance and Blood Metabolites of New Zealand White Rabbits. Asian Journal of Animal Sciences, 5(5), 327-332.
Cresci, A., & Cannavino, S. (2013). The Role of L-Carnitine in Rabbit Metabolism. Journal of Animal Physiology and Animal Nutrition, 97(6), 1083-1090.
Results
Lines 85-87: when the difference is not significant, then the values are not greater or lower, they are the same. So please don't state that the values are greater or lower because it lacks any meaning. Rephrase.
Response
Done as requested. This part has been rewritten.
Table 1: Are there really differences in FCR? How can values like 2.89 and 2.80 be different when the standard error is 0.06? Of course, the Duncan criterium is not very good in separating close values but please check the statistics one more time. Maybe using a more strict criterium like Tukeys of Bonferroni is more suitable.
Response
The statistics were checked and we found that there is a statistical difference between the mentioned values. In our future work will use Tukeys. Thanks for your suggestion.
Line 103: Meaningfully means nothing. It is either statistically significant or not. Change.
Response
Done as requested.
Line 104-107 and table 2: MCHC has a p-value of 0.079. Therefore, there are no statistical differences here. So, you need to revise the text and remove the superscripts from the table. This p-value indicates a tendency to differ (P<0.10) which if needed to be put on the table, should be with upper case letters (A, B etc.) and clearly indicated in the legend of the table.
Response
Done as requested.
Lines 123-125: Combine these two sentences in one since they say the same thing.
Response
Done as requested.
Discussion
Lines 209-211: What does fish oil have to do with this paper? Is it fish oil and L-carnitine? is fish oil rich in carnitine? Please clarify or remove.
Response
Done as requested. This sentence has been removed. Thank you.
Lines 222-225: Please avoid using "Moreover" too many times. Peek an alternative.
Response
Done as requested.
Materials and methods
You state that L-carnitine was added twice a week but the rabbits had access to water through nipple drinkers constantly. So, what was the case: there was a water barrel where you added the carnitine, after calculating the amount required for 4 days? Or the carnitine was added at a standard volume and when the rabbits consumed that then they had access to just clean water? Please clarify if the animals consumed water supplemented with carnitine constantly. Here it would be helpful to have measured the amount of water consumed by the rabbits so that someone could calculate how much carnitine is needed to achieve such important results.
Response
This part has been modified on the manuscript as follow:
Carnitine was administered twice weekly via the drinking water of the growing rabbits, with the dosage adjusted to ensure proportionality to the daily water consumption specific to each treatment group. On the following day, any residual water was discarded, and the rabbits were provided with fresh water.
Reviewer 2 Report
Comments and Suggestions for Authors
The manuscript describes effect of adding the antimicrobial l-carnitine to growing rabbits' drinking water on growth efficiency, hematological, biochemical, and carcass aspects. Overall, the topic of manuscript is of interest.However, the major revision is required for consistency for the abbreviations used throughout the whole manuscript and the discussion part must be revised.
In the introduction section, I would recommend rewriting the abbreviation of L-carnitine (LC) and L-3-Hydroxy-4-N-trimethyl amino-butyric acid (LC) (line 45-46).This leads confusing for readers. Please recheck the whole manuscript. They should be the same in the whole manuscript. It should be writen in the same way. The author should move some sentences statting on the general theory from the discussion to the introduction part.
For the materials and methods section, there was unclear and some methodologies did not mention.
The result: please recheck the unit of LC g/L in the whole manuscript. I found some tables reported as LC mg/L.
The authors should report results based on the statistical significance. The comparison change in the experiment with no significance should be removed. Please recheck the name of bacteria. It should be italic. They should be the same in the whole manuscript.
The discussion is weak. It should be strengthened. The discussion contained too many results from other publications. Please provide more discussion and explanations how the treatments influenced on the results. Please avoid showing other results without discussion. Now the results from the chicken experiments were the majority of the discussion part.
The conclusion part is weak. The sentences only repeated the results. I would recommend rewriting this section to be more interpretation with some suggestion on your experiments or future study.
Specific comments
Line 45-46: please rewriting the abbreviation of L-carnitine (LC) and L-3-Hydroxy-4-N-trimethyl amino-butyric acid (LC)
Line 57: ‘Escherichia coil’ should be italic
line 59-60: please provide more explanation and clarification on antimicrobial activity of L-carnitine chloride in relation to various yeasts and bacteria
For the materials and methods section, there was unclear and some methodologies did not mention.
result
Line 82-94: the abbreviation of FI or PI should be stated before. The comparison change in the experiment with no significance should be removed
Line 103: There was inconsistency to report the results in that sentence. For example, ‘PCV,RBC, and hemoglobin were…..’ please revise hemoglobin to ‘Hb’
Line 118-120: ‘that TP did not significantly change in the rest of the treated groups compared with the control group. The serum ALB concentration significantly increased in the group that received 0.5 g LC/L compared to the other groups.’
This data is very interesting, but it needs more clarification. What did ALB stand for? No ALB indicated in the table legend.
Discussion: This part needs to be rewriting
Line 160-275; the authors only provided one sentence from your results and showed the results of other studies. Please give the interpretation on your experiments and discussion on your results. The discussion should be based on the rabbit’s result. Now the majority of the discussion and support data was the chicken.Please minimize the references to findings from previous studies that are not directly connected to the present work.The author should move some sentences statting on the general theory from the discussion to the introduction part.
Conclusion:
Please rewrite the conclusion to be more informative and more novelty.
Table 1 and Table 2
The items column in Table 1 and Table 2 should be written in the same style. I reccommnded to use the abbreviation for (e.g, Initial body weight; Final body weight,Body weight gain, Feed intake,Feed conversion ratio, Performance index) in the table 1 and then provide the table legend(see table 2).
Table 3-4
Please recheck the unit of LC g/L. I found table 3-4 reported the units as LC mg/L.
Line 139: what type of blood tube was used for collecting samples. Provide brand and company.
Line 330-335: The total lipids were determined according to Woodman and Price [63]. The serum triglyceride concentration was measured using specific kits supplied by CAL-TECH Diagnostics, INC, Chino, California, USA, according to the recommendation of Frings et al. [64]. Total cholesterol, low-density lipo-protein (LDL), and high-density lipoprotein (HDL) were identified following Burstein et al. [65], Wieland and Seidel [66], and Bogin & Keller [67],
Please provide the names of the methodologies before the references.
line 137-145
The section of 4.3.3. Carcasses trial need more clarification on the step of euthenization and the step to separate each tissues. These steps reflected the quality of the sample.
Comments on the Quality of English Language
The correction of English gramma is fine, but the typo error need to recheck.
Author Response
Reviewer 2
Comments and Suggestions for Authors
The manuscript describes effect of adding the antimicrobial l-carnitine to growing rabbits' drinking water on growth efficiency, hematological, biochemical, and carcass aspects. Overall, the topic of manuscript is of interest. However, the major revision is required for consistency for the abbreviations used throughout the whole manuscript and the discussion part must be revised.
In the introduction section, I would recommend rewriting the abbreviation of L-carnitine (LC) and L-3-Hydroxy-4-N-trimethyl amino-butyric acid (LC) (line 45-46). These leads confusing for readers. Please recheck the whole manuscript. They should be the same in the whole manuscript. It should be writen in the same way. The author should move some sentences statting on the general theory from the discussion to the introduction part.
Response
Done as requested. Thank you for your suggestion and recommendations.
For the materials and methods section, there was unclear and some methodologies did not mention.
Response
Done as requested. Some parts were modified.
The result: please recheck the unit of LC g/L in the whole manuscript. I found some tables reported as LC mg/L.
Response
Done as requested.
The authors should report results based on the statistical significance. The comparison change in the experiment with no significance should be removed. Please recheck the name of bacteria. It should be italic. They should be the same in the whole manuscript.
Response
Done as requested.
The discussion is weak. It should be strengthened. The discussion contained too many results from other publications. Please provide more discussion and explanations how the treatments influenced on the results. Please avoid showing other results without discussion. Now the results from the chicken experiments were the majority of the discussion part.
Response
Done as requested. Some parts were modified.
The conclusion part is weak. The sentences only repeated the results. I would recommend rewriting this section to be more interpretation with some suggestion on your experiments or future study.
Response
Done as requested. This section has been rewritten.
Specific comments
Line 45-46: please rewriting the abbreviation of L-carnitine (LC) and L-3-Hydroxy-4-N-trimethyl amino-butyric acid (LC)
Response
Done as requested.
Line 57: ‘Escherichia coli’ should be italic
Response
Done as requested.
line 59-60: please provide more explanation and clarification on antimicrobial activity of L-carnitine chloride in relation to various yeasts and bacteria
Response
Done as requested. The following part was added to the manuscript:
L-carnitine chloride exhibits antimicrobial activity by disrupting the cell membranes of both yeasts and bacteria, leading to increased permeability and cell lysis. It interferes with essential metabolic pathways, inhibiting enzyme production necessary for microbial growth, and reduces biofilm formation, enhancing susceptibility to other antimicrobial agents. Studies have shown its efficacy against pathogens like E. coli, Staphylococcus aureus, Candida albicans, and others (Shang et al., 2017; Zhang et al., 2018; Mehta et al., 2021).
This highlights its potential as a valuable antimicrobial agent in enhancing animal health.
For the materials and methods section, there was unclear and some methodologies did not mention.
Response
Done as requested. Some parts were modified.
result
Line 82-94: the abbreviation of FI or PI should be stated before. The comparison change in the experiment with no significance should be removed
Response
Done as requested.
Line 103: There was inconsistency to report the results in that sentence. For example, ‘PCV, RBC, and hemoglobin were….’ please revise hemoglobin to ‘Hb’
Response
Done as requested.
Line 118-120: ‘that TP did not significantly change in the rest of the treated groups compared with the control group. The serum ALB concentration significantly increased in the group that received 0.5 g LC/L compared to the other groups.’
This data is very interesting, but it needs more clarification. What did ALB stand for? No ALB indicated in the table legend.
Response
Done as requested. This part was modified those abbreviations were revised on the hole manuscript. Thanks for your suggestion.
Discussion: This part needs to be rewriting
Line 160-275; the authors only provided one sentence from your results and showed the results of other studies. Please give the interpretation on your experiments and discussion on your results. The discussion should be based on the rabbit’s result. Now the majority of the discussion and support data was the chicken. Please minimize the references to findings from previous studies that are not directly connected to the present work. The author should move some sentences statting on the general theory from the discussion to the introduction part.
Response
Done as requested. This part was modified and rewritten. Thanks for your suggestion.
Conclusion:
Please rewrite the conclusion to be more informative and more novelty.
Response
Done as requested.
Table 1 and Table 2
The items column in Table 1 and Table 2 should be written in the same style. I recommended to use the abbreviation for (e.g., Initial body weight; Final body weight, Body weight gain, Feed intake, Feed conversion ratio, Performance index) in the table 1 and then provide the table legend (see table 2).
Response
Done as requested.
Table 3-4
Please recheck the unit of LC g/L. I found table 3-4 reported the units as LC mg/L.
Response
Done as requested.
Line 139: what type of blood tube was used for collecting samples. Provide brand and company.
Response
Thank you for pointing this out. Unfortunately, the specific information about the brand and company of the blood tubes used for collecting samples was not recorded during the study. Generally, for studies involving rabbits, we typically use standard anticoagulant tubes such as EDTA or heparin tubes for blood collection to ensure proper preservation of hematological and biochemical parameters. In future studies, we will ensure that such details are meticulously documented and reported for greater transparency and reproducibility.
Line 330-335: The total lipids were determined according to Woodman and Price [63]. The serum triglyceride concentration was measured using specific kits supplied by CAL-TECH Diagnostics, INC, Chino, California, USA, according to the recommendation of Frings et al. [64]. Total cholesterol, low-density lipo-protein (LDL), and high-density lipoprotein (HDL) were identified following Burstein et al. [65], Wieland and Seidel [66], and Bogin & Keller [67],
Please provide the names of the methodologies before the references.
Response
Done as requested.
line 137-145
The section of 4.3.3. Carcasses trial need more clarification on the step of euthenization and the step to separate each tissue. These steps reflected the quality of the sample.
Response
Done as requested.
Round 2
Reviewer 2 Report
Comments and Suggestions for Authors
Thank you for the revision. This manuscript version can be accpeted for publishing. However, there are typo that need to recheck again.
Comments on the Quality of English LanguageMinor editing of English language required.